# An assessment of risk factors for contracting rabies among dog bite cases recorded in Ward 30, Murewa district, Zimbabwe

Enica Chikanya[1,2], Margaret Macherera[3]*, Auther Maviza[2]

**1** Ministry of Health and Child Care, Seke, Zimbabwe, **2** National University of Science and Technology, Faculty of Applied Science, Department of Environmental Science and Health, Bulawayo, Zimbabwe, **3** Lupane State University, Faculty of Agricultural Sciences, Department of Crop and Soil Sciences, Lupane, Zimbabwe

* mmacherera@lsu.ac.zw

## Abstract

### Background

Zoonoses are a major threat to human health. Worldwide, rabies is responsible for approximately 59 000 deaths annually. In Zimbabwe, rabies is one of the top 5 priority diseases and it is notifiable. It is estimated that rabies causes 410 human deaths per year in the country. Murewa district recorded 938 dog bite cases and 4 suspected rabies deaths between January 2017 and July 2018, overshooting the threshold of zero rabies cases. Of the 938 dog bite cases reported in the district, 263 were reported in Ward 30 and these included all the 4 suspected rabies deaths reported in the district. This necessitated a study to assess risk factors for contracting rabies in Ward 30, Murewa.

### Methodology/ Principal findings

A descriptive cross sectional survey was used for a retrospective analysis of a group of dog bite cases reported at Murewa Hospital, in Ward 30. Purposive sampling was used to select dog bite cases and snowball sampling was used to locate unvaccinated dogs and areas with jackal presence. The dog bite cases and relatives of rabies cases were interviewed using a piloted interviewer-administered questionnaire. Geographical Positioning System (GPS) coordinates of dog bite cases, vaccinated and unvaccinated dogs and jackal presence were collected using handheld GPS device. QGIS software was used to spatially analyse and map them. Dog owners were 10 times more likely to contract rabies compared to non-dog owners (RR = 10, 95% CI 1.06–93.7). Owners of unvaccinated dogs were 5 times more likely to contract rabies compared to owners of vaccinated dogs *(RR = 5.01, 95% CI 0.53–47.31)*. Residents of the high density cluster (area with low cost houses and stand size of 300 square meters and below) were 64 times more likely to contract rabies compared to non-high density cluster residents *(RR = 64.87, 95% CI 3.6039–1167.82)*. Participants who were not knowledgeable were 0.07 times more likely to contract rabies, compared to those who had knowledge about rabies. *(RR = 0.07, 95% CI 0.004–1.25)*.

**Data Availability Statement:** All relevant data are within the manuscript and its Supporting Information files.

**Funding:** The authors received no specific funding for this work.

**Competing interests:** The authors have declared that no competing interests exist.

Our study shows that the risk factors for contacting rabies included; low knowledge levels regarding rabies, dog ownership residing in the high density cluster, owning unvaccinated dogs and spatial overlap of jackal presence with unvaccinated dogs.

## Author summary

Worldwide, rabies, a neglected tropical disease from ancient times is responsible for an estimated 59 000 human deaths a year. Between January 2017 and July 2018, an outbreak of human rabies in Murewa district Ward 30 prompted us to assess risk factors for contracting human rabies in the affected area. We reviewed cases of human rabies and dog bites through interviews and mapping of dog bite cases, vaccination status of dogs and jackal presence in the affected neighbourhood. We found a total of 263 dog bite cases including 4 suspected human rabies deaths within one year in a retrospective review of records in Ward 30 of Murewa district. Dog bites were responsible for all the rabies cases. Dog ownership, non-vaccination of dogs, residence in the high density cluster and poor knowledge about rabies were significantly associated with contracting rabies. We concluded that there was high proportion (74.86%) of low knowledge levels regarding rabies which could be a risk factor for rabies, dog ownership and non-vaccination of dogs are practices that may expose individuals to rabies, residence in the high density cluster is a risk factor for contracting rabies, unvaccinated dogs in Ward 30 are a potential risk factor for contracting rabies vis-à-vis the distribution of dog bites and spatial overlap of jackal presence, unvaccinated dogs and dog bite cases is a risk factor for rabies. We recommended intensified health education efforts on rabies by health workers in Ward 30, use of the One Health approach by various stakeholders in the district, intensified, regular mass dog vaccination campaigns by the Veterinary Department in light of jackals' presence in the area and further studies to assess dog bite and rabies management.

## Introduction

Zoonotic diseases are a major global threat to human health and sustainable development [1]. Rabies is a public health problem from ancient times and it is currently responsible for an estimated 59 000 human deaths a year, almost all transmitted via dog bites [2]. Rabies has no cure, but it is preventable with prophylaxis [3] and by the time of clinical onset it is invariably fatal [3]. More than 95% of deaths occur in Africa and Asia, 80% of which are in people living in rural areas, underserved populations; most of whom are children [3]. The jackal is considered the most important reservoir host of Rabies virus (RABV) in Zimbabwe [4]. Jackals infect dogs and may initiate outbreaks of rabies in dogs [5]. The risk of rabies spread due to jackal presence is supported by other studies [6], where they found that it is highly feasible for a foraging jackal to track resources within its home range, perhaps using prior knowledge of locations [6]. There is a global framework to eliminate human deaths from dog-mediated disease by 2030 [7]. In order to achieve this, risk factors for contracting rabies require special attention. A risk factor is any attribute, characteristic or exposure of an individual that increases the likelihood of developing a disease or injury [7]. In Zimbabwe, on average 150 animal cases of rabies are reported every year [8]. In humans, deaths from rabies have increased from two deaths in 2010 to sixteen in 2014 [8]. Rabies deaths have been reported in Mashonaland East Province (Seke, Murewa and Wedza Districts), Mashonaland West Province (Chegutu, Kadoma and

Hurungwe Districts), Mashonaland Central Province (Mt Darwin and Mazowe Districts), Midlands Province (Gokwe District), Masvingo Province (Bikita, Chiredzi, Gutu, Zaka, Mwenezi Districts) and Manicaland Province (Makoni, Mutare and Mutasa Districts), which have been ranked as rabies high risk areas [8]. Murewa District recorded a total of 938 dog bite cases and 4 rabies deaths in 2018 [9]. Of the 938 dog bite cases reported in the district, 263 were reported at Murewa Hospital in Ward 30 and these included all the 4 deaths reported in the district [9]. This is despite the fact that there are existing rabies prevention and control interventions in the district, yet there is an increase in dog bites and incidence of rabies. Geographical Information Systems (GIS) have been used to conduct studies on rabies, including those aimed at assessing risk factors for contracting the disease [10,11]. The present study sought to assess risk factors for contracting rabies in Ward 30 of Murewa District through assessment of knowledge, determining practices that may expose people to rabies, mapping dog bite hot spots, determining the vaccination status of owned dogs and determining the spatial distribution of jackals and dogs in relation to dog bite cases.

## Materials and methods

### Ethics statement

A rabies outbreak is a public health emergency which calls for appropriate outbreak response measures. The current study was a form of response to a rabies outbreak. Permission to conduct the study was obtained from the office of the Provincial Medical Director, Mashonaland East Province and the District Medical Officer, Murewa District. Consent was obtained from each participant, before conducting the interview. Consent was verbal; participants were asked if they would like to answer a few questions. They were given the choice to take or to decline the interview. To protect confidentiality, participants' names were not recorded on the questionnaire. In addition, they were assured that their responses would be kept secret. Clearance was also sought from the National University of Science and Technology.

### Study area

Murewa district is located in Mashonaland East province in north eastern Zimbabwe. It shares its boundaries with Goromonzi, Mutoko, Marondera, Murewa and Uzumba Maramba Pfungwe (UMP) districts. The district, according to the 2012 census, has a population of195 085 with 93 367 males and 101 718 females [12]. It has 28 wards which are wholly communal and two wards which are Growth points. Growth points, in the Zimbabwean context, are settlements which central and local government consider having potential for development and needing to be supported by further public and private sector investment [12]. Murewa falls under Natural Region 2. This region is located in the middle of the north of the country. The rainfall ranges from 750 to 100 millimetres per year [13]. The rainfall is fairly reliable, falling from November to March/April. Because of the reliable rainfall and generally good soils, Natural Region 2 is suitable for intensive cropping and livestock production. It accounts for 75–80 percent of the area planted to crops in Zimbabwe [13]. The cropping systems are based on flue-cured tobacco, maize, cotton, wheat, soybeans, sorghum, groundnuts, seed maize and burley tobacco grown under dry land production as well as with supplementary irrigation in the wet months. Natural Region 2 is suitable for intensive livestock production [13]. Prior to 2000, the region was dominated by large-scale farming subsector characterised by highly mechanised farms of 1000–2000 hectares under freehold title and owner-operated. Following the agrarian and land reform programmes initiated in 1999/2000, a large proportion of the farms were subdivided into smaller units and allocated to new farmers under the A1 and A2 small-scale farming system [13]. The A1 model allocated small plots to for growing crops and

grazing land to landless and poor farmers, while the A2 model allocated farms to new black commercial farmers who had the skills and resources to farm profitably, reinvest and raise agricultural productivity [14]. Fig 1 shows the study area map.

As shown in Fig 1, the central part of the ward (represented by the light grey, pale yellow and light blue coloured features on the Ward 30 main map image) consists of a growth point or central business district (CBD) of Murewa, which is developing towards achieving a town status. The infrastructure in Ward 30 includes an industrial area, various government institutions and places of residence (i.e. high, medium and low density suburbs). The map also shows that there is no development as one moves out/ away from the CBD i.e. the area is characteristically a typical bushy savannah landscape with sparsely distributed rural settlements.

The economy of the district is primarily agrarian with potential for mining especially black granite, gold and tantalite. The major farming activities are crop cultivation; livestock rearing; vegetable and dairy production. Most of the farming is subsistence. Other economic activities include mining and tourism [14]. Other activities include horticulture and traditional hunting, where dogs are used to chase after prey. The district has 26 clinics and 3 hospitals. Veterinary

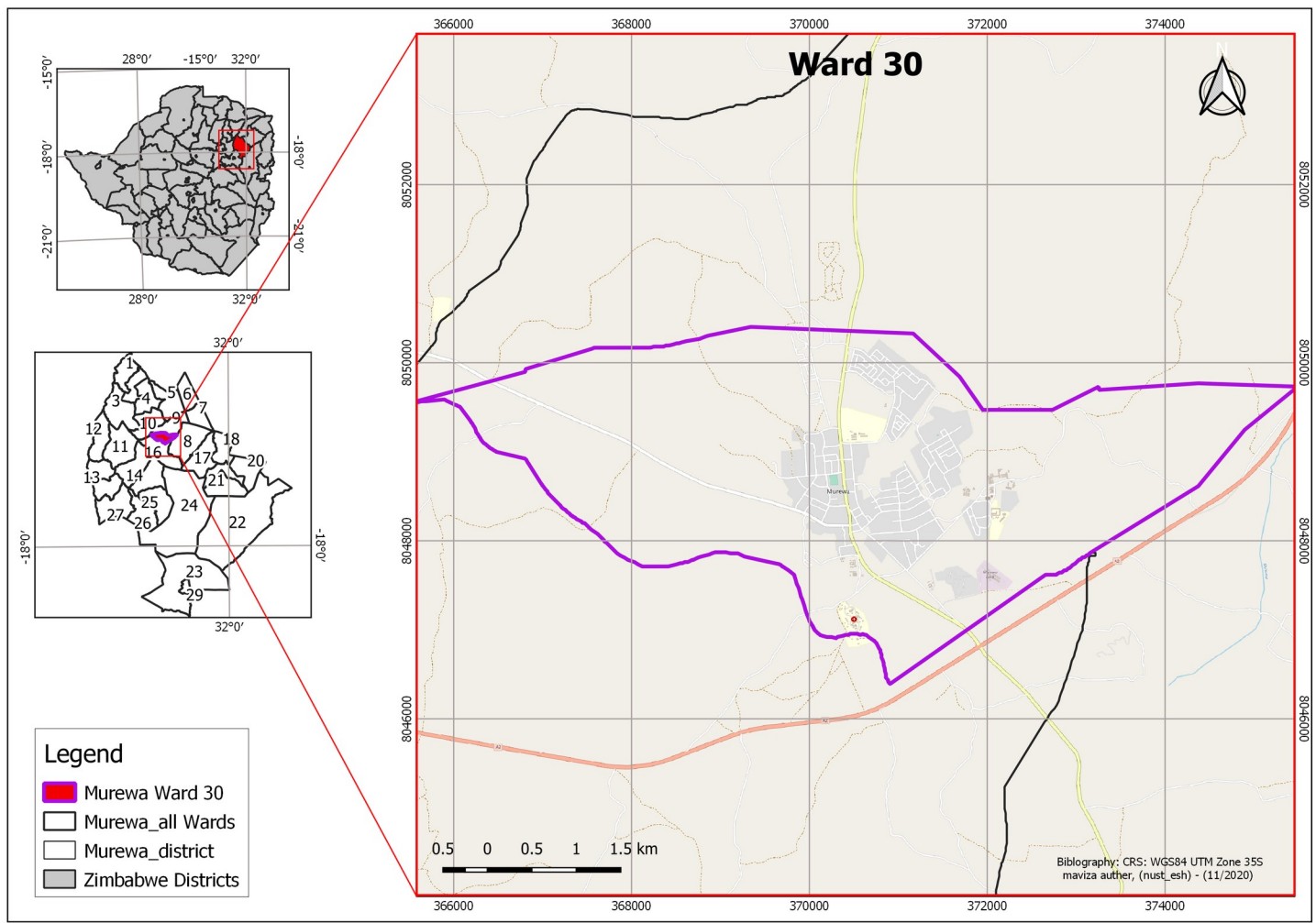

**Fig 1. Map of the study area area Created by A Maviza (Ward 30, Murewa District, Zimbabwe).**

Services Department and Zimbabwe Republic Police are among many other services available in the area [14].

## Methods

Across sectional descriptive survey was conducted in Ward 30, Murewa District. This study type was chosen on the basis that it allowed for retrospective analysis of dog bite and rabies cases that were reported in between January 2017 and July 2018. The target population were residents of Murewa District, jackals and dogs in the district. The study was guided by the variables shown in the summary of variables (S1 Text). Multistage sampling was conducted. This involved purposive sampling to select Murewa district which is the district with a high number of dog bite and rabies cases in the province. Ward 30 was also purposively sampled because it reported the highest number of dog bites and human rabies cases, in the district. A total sample of all the 263 dog bite cases recorded in Murewa Hospital were enrolled for the study.

A total of 263 dog bite cases were obtained from the Murewa hospital register in Ward 30. Murewa hospital is the health facility that serves Ward 30. However, some patients from outside Ward 30 may choose to utilise the hospital and will be recorded in the Murewa Hospital register. This number included 4 suspected human rabies cases. Sixty eight dog bite cases out of the 263 could not be reached for interview. The total number of study participants was thus 195. The 195 participants included relatives of the 4 deceased suspected rabies cases. One hundred and ninety one (191) dog bite cases were interviewed and one family member from each household that had a rabies case was selected for a proxy interview using a questionnaire. The age range for the proxy interviewees was 19–59. The enrolees were recruited by virtue of them being recorded as dog bite cases in the Murewa hospital register for the period January 2017 to July 2018 and all dog bites and rabies cases recorded during this period were enrolled for the study. The local language (ChiShona) was used for the interviews. Both men and women were interviewed as long as they were dog bite cases or relatives of the deceased suspected rabies cases. Surveys were done using printed questionnaires for interviews.

Snowball sampling was employed to select the desired dogs and jackals (presence) samples. Owners of vaccinated and unvaccinated dogs in Ward 30 were tracked and georeferenced. Snowball sampling was used because the total number of the dogs in the ward was not known. Owners of vaccinated dogs were located through records at the Murewa District Veterinary Offices records while those with unvaccinated dogs were located through the snowball sampling approach whereby, a sampled dog owner (regardless of dog vaccination status) was asked to help us locate the nearest dog owner(s) they knew within their respective localities. Residents residing at the peripheral households of the settlement area were sampled and data regarding the presence of jackals near their areas of residence solicited for. The data collected included location points (in geographic coordinates) of eye-witness accounts/ sightings of jackals. Furthermore, location points of jackal spoor and scat evidence in the area was sought. The jackal spoor and scat were identified using the Scat method adopted from Mukherjee [15]. This method involved conventional wildlife, field scat-identification using common characteristics such as morphology (i.e. diameter, length and shape), colour, odour and physical appearance. The location points for spoors were also identified using common wildlife canine characterisations such as heel pad and lobes size, outer and inner toes, outer and inner track size and orientation and claw size. This field activity was carried out with assistance from two animal tracking experts from the Zimbabwe National Parks and Wildlife Authority (ZNPWA). The geographic coordinates of the reported jackal sightings and identified spoors and scat were captured using a Garmin etrex30 handheld GPS device (in latitude, longitude format).

Location of vaccinated dogs, unvaccinated dogs and dog bites were also captured using the GPS handheld device. A piloted interviewer-administered questionnaire designed using guidance from previous studies [16,17] was used to assess knowledge and practices pertaining to rabies from dog bite cases and relatives of the suspected rabies cases. The questionnaire was corrected after the piloting which involved 34 participants. Improvements were made on knowledge questions by clearing ambiguities. The questionnaire was translated to Chishona the local language. One researcher translated from English to Chishona and the other researcher translated it back to English to check if the meaning remained the same. Knowledge questions were to do with the animal reservoir of rabies in Zimbabwe, signs and symptoms, methods of transmission of rabies, prevention of rabies, treatment seeking following a dog bite and rabies vaccination in dogs. Practice topics encompassed time taken to seek treatment following a dog bite, where to seek treatment, dog ownership and vaccination status of owned dogs. The overall content also included demographic characteristics, dog bite hotspots and spatial distribution of jackals in relation to rabies cases.

The questionnaire took an average time of 30 minutes to administer. A data capture form was used to record captured GPS coordinates.

## Statistical analysis

Before analysis, data were cleaned for errors. Quantitative, qualitative and spatial data were generated. The questionnaire responses on knowledge and practices regarding rabies were analysed using Epi Info 7 statistical package to generate frequencies and percentages. The Likert Scale, in conjunction with a scalar scoring method (high; medium; low) adapted from another study [18] was used to score and rate responses to knowledge questions. A Likert Scale is the most widely used psychometric approach to ask study participants about their opinion in survey research using usually 5 or 7 answer options range. The participants can give a negative, neutral or positive response to a statement. They are usually used to gauge agreement, importance or likelihood. Risk ratio was calculated to determine the relationship between dog ownership and contracting rabies. Microsoft Excel was used for cleaning, correction and presentation of qualitative and quantitative data in the form of tables and graphs.

## Spatial analysis

Spatial data analysis entailed the following main steps: (1) data pre-processing (2) attribute data appendage to spatial data, (3) spatial overlaying and proximity analysis and (4) geovisualisation using Quantum GIS (QGIS) 3.2.2 software. All captured coordinate data (in Latitude, Longitude format) were downloaded from the GPS device and uploaded into QGIS and then converted into vector format (shapefiles). These were then reprojected from World Geodetic System of 1984 (WGS84) to Universal Transverse Mercator (UTM) Zone 35South (X, Y) projection to allow for use of metric spatial measurement units (metres) during proximity analysis and efficient overlaying of all shapefiles i.e. jackal presence, dog vaccination status and dog bite cases in Ward 30.

Dog bite cases, vaccination status and jackal presence shapefiles were overlaid over the Open Street Map background layer and different symbology used to distinguish, visualise and map the different aspects under study. The Inverse Distance Weighting (IDW) interpolation algorithm [11] within the QGIS spatial toolbox was then used to generate hotspot maps for jackal presence, dog vaccination status and dog bites whilst proximity analysis using the buffering technique was applied on the jackal presence shapefile (using a 12km buffer range for jackals' travel range [19]). This was done so as to determine and visualise the potential spatial overlaps with dog bites and vaccination status. Furthermore, spatial relationships/ associations

were then depicted in the overlaying process of the hotspot maps and the other mapped aspects of the study (i.e. vaccination status, dog bites and rabies cases). Standard map cosmetics were then applied i.e. main map elements (e.g. North arrow, legend and scale) to complete and export the maps in JPEG format. Final outputs maps (refer to results section) included a dog bite, dog vaccination status, jackal presence and range maps and arisk map showing potential spatial interaction between dog bite cases, unvaccinated dogs and jackals.

## Results

### Socio-demographic characteristics of participants

Socio demographic characteristics of the participants are shown in Table 1. A total of 263 enrolees were enrolled into this study. However, 195 of them were located while the remaining 68 could not be found at addresses that were recorded in the Dog bite register at Murewa Hospital. The group of participants was male dominated, where such participants constituted 68.20% of the group. Most of the participants were aged between 12 and 39 years of age (69.74%).

Participants who attained secondary level education, which is an attainment of at least eleven years in school, where 7 years of Primary Education are first completed before proceeding to 4 years of secondary school or in some instances, 6 years (2 additional years of Advanced secondary education), constituted 61.03%, while only 9.23% reported to have completed tertiary education. Primary Education was attained by 25.13% of the participants while the remaining 4.62% reported that they had never attended school. Christianity was the religion of most of the participants (58.46%). Apostolic Sect members constituted 27.18% of the participants.

### Knowledge and practices regarding rabies

An assessment of knowledge regarding rabies revealed that 74.86% of the 195 participants had a low level of knowledge. The participants comprised 191 dog bite cases and 4 relatives of the

**Table 1. Socio-demographic characteristics of study participants (n = 195).**

| Variable | Frequency (n) | Percentage (%) |
|---|---|---|
| Sex | | |
| Male | 133 | 68.20 |
| Female | 62 | 32.46 |
| Age group | | |
| 6–11 | 17 | 8.72 |
| 12–19 | 51 | 26.15 |
| 20–29 | 41 | 21.03 |
| 30–39 | 44 | 22.56 |
| 40–49 | 19 | 8.90 |
| 50–59 | 11 | 5.64 |
| 60+ | 12 | 6.15 |
| Level of education | | |
| None | 9 | 4.62 |
| Primary | 49 | 25.13 |
| Secondary | 119 | 61.03 |
| Tertiary | 18 | 9.23 |
| Religion | | |
| Christianity | 114 | 58.46 |
| Apostolic sect | 53 | 27.74 |
| Muslim | 19 | 9.74 |
| Traditional | 7 | 3.59 |
| Other | 2 | 1.03 |

deceased rabies cases. Only 17.8% had a high knowledge level on rabies. Dog ownership, non-vaccination of owned dogs, time taken to seek treatment following a dog bite and places of treatment were the assessed practices. Dog ownership was reported by 76.92% of the study participants. Among the dog owners, 75.55% had their dogs vaccinated. Following a dog bite, 24.35% of the participants said they sought treatment immediately while most (52.85%) reported to have sought treatment within a week following a dog bite. Some of them (21%) of sought treatment within a month. Findings also revealed that most of the dog bite cases (96.4%) sought treatment at health facilities, while some (1.53%) opted for treatment from traditional healers and the remainder (0.51%) opted for other options. Detailed results are shown in Table 2.

Table 3 shows the comparison between the relatives of rabies cases as a proxy to the rabies cases and dog bite cases in terms of knowledge, practices and area of residents. The associations could not be tested due to the low number of actual rabies cases. The results however show that lack of knowledge about rabies may have an association with contracting rabies. Practices like time taken to visit the clinic after a dog bite, dog vaccination and owning a dog may also have an association with contracting rabies. Another factor that may be associated with contracting rabies is living in the high density areas and having observed jackal presence. The following are examples of the questions that were asked to all participants 'What is the animal reservoir of rabies in Zimbabwe?' What are the signs and symptoms of rabies in dogs? And 'What are the methods of rabies transmission?' A total of 191 dog bite cases and 4 relatives of the deceased rabies cases were interviewed (n = 195).

There was evidence of major association between dog ownership and contracting rabies. Results show that dog owners were more likely to contract rabies as compared to non-dog owners (RR = 10, 95% CI 1.06–93.7). Owners of unvaccinated dogs were more likely to contract rabies as compared to owners of vaccinated dogs (RR = 5.01, 95% CI 0.53–47.31). See Table 4.

## Dog bite and vaccination status spatial distribution and hotspots

**Dog bites.**   Results revealed most (96%) of dog bites in Ward 30 were reported in the High Density Cluster. Only 0.51% was reported in the low density area. The remaining 3.49% cases were noted to be outside Ward 30catchment area, though they were reported at Murewa Hospital, which is in Ward 30. Fig 2A further illustrates spatial distribution of the 24 dog bites in Ward 30 with 2 distinct hot spots in the northern extent and the central region of the Ward (see Fig 2B) though there are most cases (83.3%) are situated on the eastern side of the major road. Residents of the High Density Cluster were more likely to contract rabies as compared to residents from other areas RR = 64.87, 95%CI 3.6039–1167.82 (p<0.05). This is supported by the revealed hot spots and spatial clustering of dog bites which are mostly in the high density cluster.

**Vaccination status.**   Findings on the vaccination status of dogs in Ward 30 are shown in Fig 2C and 2D. The map (Fig 2C) shows spatial distribution of vaccinated and unvaccinated dogs while 2D shows hotspots for the unvaccinated dogs in the study area. Blue dots represent vaccinated dogs while the yellow dots represent unvaccinated dogs. A total of 290 dog vaccination conditions were geocoded and mapped. Two hundred and fifty-one (86.55%) of these were vaccinated dogs and 39 (13.44%) were unvaccinated dogs. The maps show possible interaction between vaccinated dogs and unvaccinated dogs showing potential for rabies transmission. The spatial distribution of the dogs basically follows the human settlement pattern in the study area though there is clustering (high concentration) of dogs in the central part of the study area. Three (3) distinct hotspots of unvaccinated dogs are revealed in this study: one in the northern extent and 2 in the south-eastern part of Ward 30 (refer to Fig 2D) showing higher risk of possible rabies transmission in these areas.

**Table 2. Responses to the questionnaire.**

| KNOWLEDGE | | | |
|---|---|---|---|
| Question/ Thematic area | Responses | Number of participants who gave the responses | Percentage (%) of participants who gavethe responses |
| *Animal reservoir of rabies in Zimbabwe* | Dog | 146 | 74.87 |
| | Jackal | 49 | 25.13 |
| | Total | 195 | 100 |
| *Signs and symptoms of rabies in dogs* | Biting without provocation | 44 | 22.56 |
| | Agitated behaviour | 55 | 28.20 |
| | Growling | 34 | 17.43 |
| | Foaming at the mouth | 32 | 16.41 |
| | Refusal of food | 25 | 12.82 |
| | Other | 5 | 2.58 |
| | Total | 195 | 100 |
| *Transmission of rabies* | Bites | 178 | 91.28 |
| | Scratches | 15 | 7.69 |
| | Open wound licking | 2 | 1.03 |
| | Total | 195 | 100 |
| *Possibility of preventing rabies* | Yes | 184 | 94.36 |
| | No | 11 | 5.64 |
| | Total | 195 | 100 |
| *Ways of preventing rabies* | Dog vaccination | 82 | 42.05 |
| | Human vaccination | 89 | 45.64 |
| | Other | 24 | 12.31 |
| | Total | 195 | 100 |
| *Knowledge of where dogs rabies vaccine can be found* | Government Veterinary Offices | 121 | 62.05 |
| | Elsewhere/ Uncertain | 74 | 37.95 |
| | Total | 195 | 100 |
| *Sources of rabies information* | Health officials/ workers | 126 | 64.61 |
| | Television | 5 | 2.56 |
| | Radio | 55 | 28.20 |
| | Internet | 4 | 2.06 |
| | Other | 5 | 2.57 |
| | Total | 195 | 100 |
| PRACTICES | | | |
| Question/ Thematic area | Responses | Number of participants who gave the responses | Percentage (%) of participants who gave the responses |
| *Time taken to visit a health facility following a dog bite* | Immediately (within an hour on the same day) | 47 | 24.10 |
| | Within a week | 102 | 52.30 |
| | Within a month | 41 | 21.03 |
| | Other | 5 | 2.57 |
| | Total | 195 | 100 |
| *Places where treatment was sought* | Nearest health facility | 188 | 96.41 |
| | Native/traditional leader | 3 | 1.53 |
| | Other | 4 | 2.06 |
| | Total | 195 | 100 |
| *Dog ownership* | Yes | 49 | 25.12 |
| | No | 146 | 74.88 |
| | Total | 195 | 100 |

*(Continued)*

**Table 2.** (Continued)

| | | | |
|---|---|---|---|
| *Vaccination status of owned dog* | Vaccinated | 35 | 17.94 |
| | Not vaccinated | 14 | 7.18 |
| | Did not own dog | 146 | 74.88 |
| | Total | 195 | 100 |
| **DOG BITE HOT SPOTS** | | | |
| Question/ Thematic area | Responses | Number of participants who gave the responses | Percentage (%) of participants who gave the responses |
| *In which part of Ward 30 were you bitten by a dog? Specify location* | High density cluster residential area | 23 | 11.79 |
| | Outside Ward 30 residential area | 172 | 88.21 |
| | Total | 195 | 100 |
| **SPATIAL DISTRIBUTION OF JACKALS** | | | |
| Question/ Thematic area | Responses | Number of participants who gave the response | Percentage (%) of participants who gave the responses |
| *Hearing or seeing jackals in the area* | Yes | 109 | 55.90 |
| | No | 86 | 44.10 |
| | Total | 195 | 100 |

## Spatial distribution of jackals, hotspots and range zone

Fig 3A shows areas where there have been jackal sightings and evidence of their presence (spoors and scat). A distinct presence hotspot (spatial clustering) of jackal presence was noted in the western extent of the Ward, while in the north-eastern side, a sparse distribution was noted (see Fig 3B). Presence of jackals (*Canis adustus)* was reported in areas surrounding the growth point and settlements, which are mainly bushy, savannah landscapes. No jackal presence was reported within the growth point and settlement areas of Ward 30. The travel range (12km) for *Canis adustus* is shown in Fig 3C. This range zone is noted to cover the entire extent of Ward 30 implying potential interaction of the jackals with dogs in the area and thus risk of rabies transmission from the jackals to the dogs.

Fig 4 is a map showing the revealed potential spatial interactions between jackals, unvaccinated dogs and reported dog bites within the 12 kilometre ranging zone of *Canis adustus* in Ward 30. These findings show a potential risk of rabies transmission not only due to likelihood of jackal-unvaccinated dog interaction but also the potential transmission to humans as evidenced by the dog bites within this same spatial extent in Ward 30. In other words, this spatial overlap of factors presents a potential high risk for rabies transmission to dogs and then to humans.

## Discussion

Identification of risk factors for contracting rabies is critical as such information may be used to raise awareness focusing on information about the risks and correct behaviours to prevent these risks. This may in turn result in the prevention of unnecessary deaths [20]. Participants of the present study had a low level of knowledge according to the adopted scoring system. The knowledge deficit was shown on critical aspects of rabies namely the reservoir of rabies in Zimbabwe and signs and symptoms of rabies in dogs. This may be attributed to the lack of utilisation of the One Health approach in dealing with zoonotic diseases in the district. Limited knowledge was found to be both a serious challenge for controlling rabies and a risk factor for human rabies, according to a similar study by [21]. Consequences of lack of knowledge on rabies are dangerous practices like domesticating unvaccinated dogs, which may expose

**Table 3. Comparison between Relatives of Rabies Cases and Dogbite cases who did not contract rabies in terms of Knowledge, practices and area of residence.**

| KNOWLEDGE ON RABIES | | | |
|---|---|---|---|
| **Variable** | **Responses** | **Relatives of Rabies Cases** | **Dog bite cases** |
| **Animal reservoir of rabies in Zimbabwe** | Jackal | | |
| | Correct | 0 (0%) | 49 (25.12%) |
| | Incorrect | 4 (2.05%) | 142 (72.82) |
| **Signs and symptoms of rabies in dogs** | Biting without provocation | 2 (1.02%) | 53 |
| | Agitated behaviour | 0 | 44 |
| | Growling | 2 | 30 |
| | Foaming at the mouth | 0 | 34 |
| | Refusal of food | 0 | 25 |
| **Methods of rabies transmission** | Biting | 4 | 174 |
| | Scratches | 0 | 15 |
| | Licking of open wound | 0 | 2 |
| **Prevention of human rabies** | Yes | 3 | 181 |
| | No | 1 | 10 |
| **Methods of rabies prevention** | Dog vaccination | 4 | 78 |
| | Human vaccination | 0 | 89 |
| **Availability of rabies vaccine at government veterinary offices** | Yes | 0 | 121 |
| | No | 0 | 0 |
| | Uncertain | 4 | 70 |
| **Source of rabies information** | Health officials | 4 | 122 |
| | Television | 0 | 5 |
| | Radio | 0 | 55 |
| **Time taken to seek treatment following a dog bite** | Immediately | 0 | 47 |
| | Within a week | 2 | 100 |
| | Within a month | 2 | 39 |
| | Other | 0 | 1 |
| **Dog ownership** | Yes | 3 | 42 |
| | No | 1 | 149 |
| **Dog vaccination status** | Vaccinated | 1 | 34 |
| | Unvaccinated | 3 | 7 |
| AREA OF RESIDENCE | | | |
| **Dog bite hotspots** | High density | 4 | 19 |
| | Medium density | 0 | 0 |
| | Low density | 0 | 1 |
| | Industrial area | 0 | 0 |
| | Institutions | 0 | 0 |
| | Central Business District (CBD) | 0 | 0 |
| | Outside catchment area | 0 | 171 |
| RESPONSES ON JACKAL PRESENCE | | | |
| **Jackal presence** | Yes | 3 | 27 |
| | No | 1 | 164 |

individuals to rabies. It is however, risky to have little knowledge on such a fatal disease as rabies because upon exposure, the effect is death due to mismanagement [3]. The finding on low level of knowledge regarding rabies in the current study is supported by other study findings. Similar studies conducted in South Asia, India, Filipinos and France also showed low

**Table 4. Risk Ratio (RR).**

| Variable | Test Group | Rabies positive (n = 4) | | Rabies negative (n = 191) | | RR & 95% CI |
|---|---|---|---|---|---|---|
| **Dog ownership** | Dog owners | 3 | 1.53% | 42 | 21.53% | 10 (1.06–93.7) |
| | Non-dog owners | 1 | 0.51% | 149 | 76.41% | |
| **Dog vaccination status** | Vaccinated | 1 | 0.51% | 121 | 62.05% | 5.01 (0.53–47.31) |
| | Unvaccinated | 3 | 0.51% | 70 | 35.89% | |
| **Dog bite hotspots** | High density cluster residents | 4 | 2.05% | 19 | 9.74% | 64.87 (3.6039–1167.82) |
| | Non- high density cluster residents | 0 | 0% | 172 | 88.2% | |

level of knowledge on rabies [22,23,24] identified a need to improve knowledge on rabies since lack of such constituted a risk factor for contracting rabies.

Study participants who did not vaccinate their dogs were more likely to contract rabies as compared to those who had their dogs vaccinated, which is supported by findings from other studies where it was proven that dog ownership and non-vaccination of dogs were risk factors for rabies outbreak initiation [23,25,26]. Vaccination of dogs is free in Zimbabwe and is

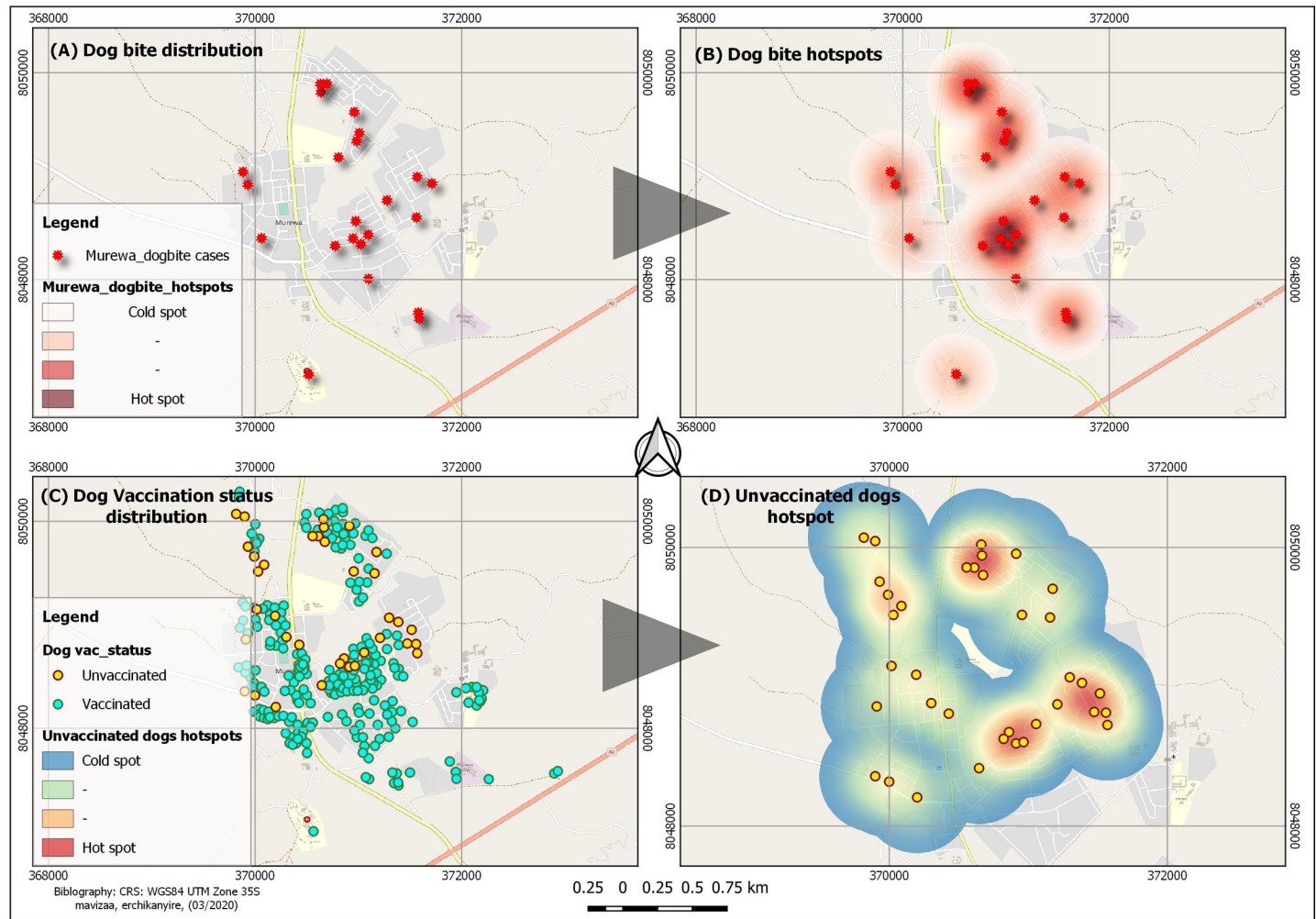

**Fig 2.** Map showing spatial distribution of dog bites (A), hotspots (B), dog vaccination status (C) and the unvaccinated dogs hotspots (D) in Ward 30, Murewa.

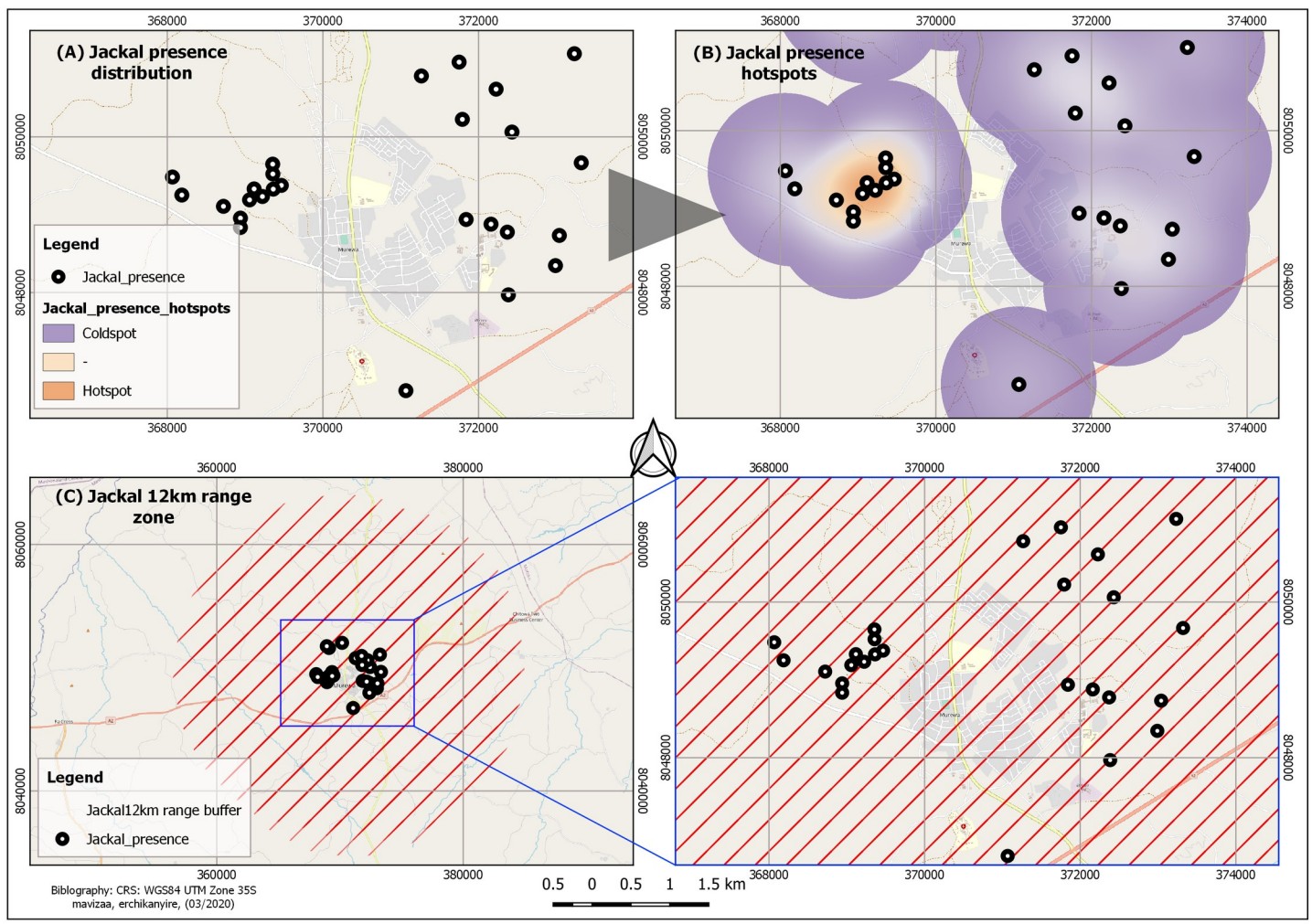

**Fig 3.** Map showing jackal presence (A), jackal presence hotspot (B), and jackal 12km range zone(C) in and around Ward 30, Murewa.

conducted by the department of Veterinary Services Department. On the other hand the administration of the vaccine to humans after exposure to rabies is very expensive. A One Health approach may help to create awareness on the importance of vaccination of dogs. Although rabies is a notifiable infectious disease in terms Zimbabwe's Public Health Act (Chapter 15:17), there are no details of priority One Health measures to focus disease prevention and control efforts. The keeping of unvaccinated dogs may also be a reflection of non-enforcement of legislation on vaccination of dogs by the Veterinary Services Department.

The layer map on dog bites (Fig 2) shows that all cases in Ward 30 were from the High Density Cluster except for one outlier, which was situated in the low density area. Results of the current study show close interaction between man and dogs in the high density cluster of Ward 30, where the majority of people reside. This may be due to the fact that people in these areas live in close proximity and if they keep dogs there is likely to be close interaction between dogs and man. The noted close interaction between man and dogs is a cause for concern and may be a risk factor for contracting rabies, as supported by findings from similar studies conducted in Tanzania, India, Zambia, Asia, Filipinos and South Africa [27,28,29].

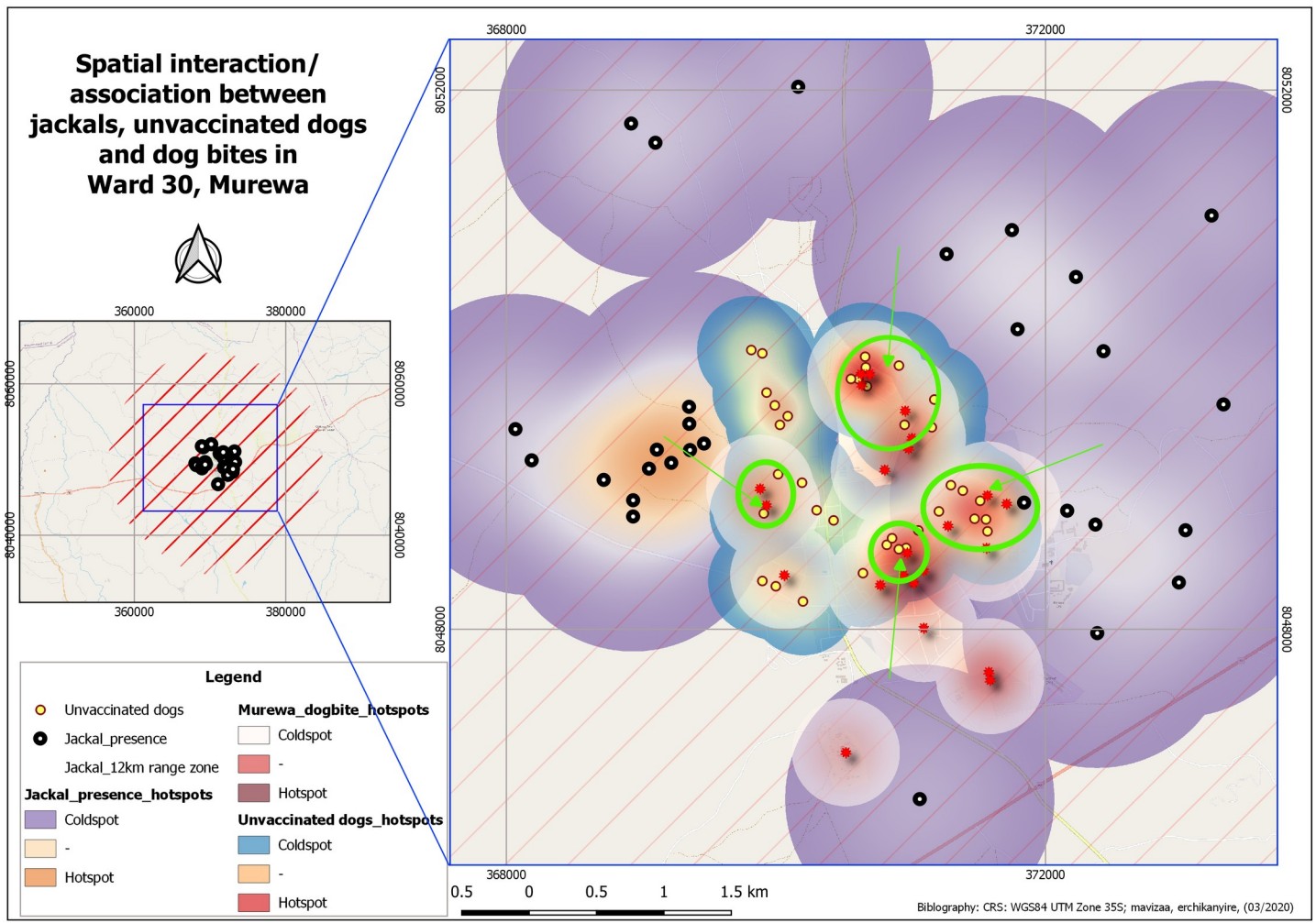

**Fig 4. Map showing spatial patterns of potential association/interaction between jackals (within 12km ranging zone) and unvaccinated dogs in relation to revealed dog bites in the study area.** (NB: Green circles and arrows depict spatial overlap between all the aspects under study).

Among the study participants, some of the dog owners had unvaccinated dogs (24.44%). The percentage of vaccinated dogs was 75.55%. The reported vaccination coverage surpassed the WHO target for dog vaccination, which is 70% [7]. However, this coverage cannot be extrapolated to the rest of the district because reports from the Murewa District Veterinary offices indicated that they had not yet met their annual target for dog vaccination and that turn out for dog vaccination was very low [4]. The low turnout for vaccination could also be a reflection of the lack of knowledge on the importance of vaccination of the dogs. Indications on the possible reasons for failure to meet vaccination targets were that the Veterinary Services Department was short of resources to carry out mass dog vaccinations, which could boost their coverage. It is evident that despite the high percentage of vaccinated dogs, unvaccinated dogs still followed human settlement patterns, which is a risk factor for contracting rabies.

It can however be deduced that the low turnout for dog vaccination may be as a result of transport costs faced by dog owners, as supported by findings from a study carried out by [30], which suggests that dog owners were more likely to present their dogs for vaccination when the vaccines were being administered in places less than a kilometre away from their homes,

where there are no transport costs involved (91%) [30]. Spatial distribution of vaccination status of dogs shows another risk factor for contracting rabies. There is interaction between vaccinated dogs and unvaccinated dogs. According to [30], unvaccinated dogs, having a high likelihood of contracting rabies from the RABV reservoir, may be able to transmit infection to vaccinated dogs whose immunity has waned, which has a potential for rabies infection and spread among the rest of the dog population, without exception of humans, since the unvaccinated dogs follow human settlement patterns. Evidence of spatial clustering of vaccinated and unvaccinated dogs was noted, which is a risk factor with potential for initiating rabies outbreaks in both the human and dog population. Spatial clustering of dog bites as a risk factor for contracting rabies is supported by results from studies carried out by other scholars [24,30,31]. The aforementioned findings on vaccination of owned dogs, together with the supporting literature from other studies, are an indication that non vaccination of dogs is a risk factor for rabies, especially when unvaccinated dogs follow human settlement patterns. Dog vaccination is an important intervention for rabies control and is a risk factor for contracting rabies if not carried out to recommended standards.

Jackal presence was reported within and outside Ward 30. Areas where jackal presence was reported within Ward 30, were outside the settlement section of the ward where it was bushy and in some instances there were rocky outcrops. All sightings and hearings of jackals were reported to be outside the area where there is human settlement, The possible reason for non-existence of jackals in such an area maybe that the area is an urban set up, as shown by the clusters in the present study (High Density, Medium Density, Low Density, Institutions, Industrial Area and the Central Business District/ Growth Point). Such an environment is not conducive for wildlife, which in many instances stays away from human settlements. The human rabies cases reported in this area may be attributed to possible interaction between jackals and dogs used for hunting by some residents in the study area. Reports of presence of jackals outside the settlement area of Ward 30 can further be supported by the landscape description of the ward's peripheries, which is suitable for jackal habitation (granite rocky outcrops; wooded savannah), which relates to studies by [6,19,32].

The 12km buffer indicating the travel range for jackals (*Canis adustus* and *Canis mesomelas*) [19] and an overlap of jackal presence, unvaccinated dogs and dog bite cases translate to possible interaction of jackals (*Canis adustus* and *Canis mesomelas*) and dogs (*Canis familiaris*). Spatial clustering of these three components is suggestive of a risk factor for contracting rabies. When unvaccinated dogs come into contact with jackals and get bitten by the wild canines, they contract rabies and in turn, bite humans, with whom they are in constant touch. The risk of rabies spread is quite high, considering that all the three aforementioned components are well within the travel range of which are the reservoirs of RABV.

Despite the impossibility of jackal presence in the area of settlement, the 12km buffer for travel range of jackals suggests possible interaction between the wildlife species (*C. adustus* and *C. mesomelas*) and domesticated dogs (*C. familiaris*), which is a risk factor for rabies. This is because the buffer encompasses both the study area (Ward 30) and areas of reported jackal sightings and hearings, which may be conclusive evidence of jackal-dog interaction, resulting in humans contracting rabies.

It is however important to note that this current study was limited by the fact that only 195 out of 263 dog bite cases were included in the study sample (due to accessibility constraints) and also that proxy interviews were conducted for the 4 rabies cases that were deceased. Furthermore, the GPS device used in this study has a geo-accuracy of ±3 metres which means that the geolocation accuracy of all the mapped data could inherently have this margin of error. The implications for these results can only be applied to people who have been bitten by a dog.

## Conclusions

In conclusion, our study shows that there was high proportion of low knowledge levels regarding rabies among the participants; dog ownership and non-vaccination of dogs are practices that may expose individuals to rabies; residence in the high density cluster is a risk factor for contracting rabies; unvaccinated dogs in Ward 30 are a potential risk factor for contracting rabies vis-à-vis the distribution of dog bites; spatial overlap of jackal presence, unvaccinated dogs and dog bite cases is an indication of a risk factor for contracting rabies.

Overall, in light of this evidence, we recommend intensified health education efforts on rabies by health workers in Ward 30, One Health approach by various stakeholders in the district and regular mass dog vaccination campaigns by the Veterinary Department in light of jackals' presence in the area.

Findings from the current study have advanced understanding of rabies through use of spatial analysis in assessment of risk factors for contracting rabies. This has brought another dimension with which such problems may be addressed. Unanswered questions which may be the basis of future studies include an in-depth analysis into dog bite management, rabies outbreak preparedness and response and feasibility studies on control of rabies in the jackal population.

## Supporting information

**S1 Text. Summary of variables.**
(DOCX)

## Acknowledgments

We thank the Ministry of Health and Child Care, through the office of the Mashonaland East Provincial Medical Director, the Murewa District Medical Officer, the National University of Science and Technology and Lupane State University. We would like to also thank the Zimbabwe National Parks and Wildlife Authority and the Veterinary Services Department for their valuable support.

## Author Contributions

**Conceptualization:** Enica Chikanya.

**Data curation:** Enica Chikanya.

**Formal analysis:** Enica Chikanya, Auther Maviza.

**Methodology:** Enica Chikanya, Margaret Macherera.

**Supervision:** Margaret Macherera, Auther Maviza.

**Validation:** Margaret Macherera, Auther Maviza.

**Writing – original draft:** Enica Chikanya.

**Writing – review & editing:** Margaret Macherera, Auther Maviza.

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
