## [Decision Letter · Decision Letter 0]

1 Oct 2020

Dear Dr macherera,

Thank you very much for submitting your manuscript "An assessment of risk factors for
contracting rabies in Ward 30, Murewa District, Zimbabwe" for consideration at PLOS
Neglected Tropical Diseases. As with all papers reviewed by the journal, your
manuscript was reviewed by members of the editorial board and by several independent
reviewers. In light of the reviews (below this email), we would like to invite the
resubmission of a significantly-revised version that takes into account the
reviewers' comments. 

The authors present an examination of risk factors for rabies in Ward 30, Murewa
District, Zimbabwe. They identify risks associated with low awareness of rabies
risks and levels of unvaccinated dogs as primary concerns. The paper provides a nice
visual spatial analysis of dog bites, jackal presence, and rabies positive cases.
There are issues with clarity in the methods that impede the interpretation of the
results that needs to be addressed. 

The methods section could use more clarity. The methods state that “one family member
from each household that had a rabies case was selected for a proxy interview using
a questionnaire”. Were these households that had human rabies cases? Elsewhere, the
authors state that there were 4 rabies positive cases. So does this mean that only 4
households were surveyed? Where then does the 191 sample size come from. Please
clarify. Some terms that were used were also confusing and inconsistently used
(e.g., respondent, participant). In the results, the authors talk about 263
respondents. Where does this number come from? Only 191 were able to respond to the
questionnaire, so the 263 individuals should not be called respondents but perhaps
‘enrollees’. How were they enrolled? What questions were asked on the questionnaire?
A table of results from these questions would help.

Understanding how the people surveyed were selected is critical for readers to
understand the implications of these results. The authors state that “dog owners
were more likely to contract rabies as compared to non-dog owners”. This makes is
sound like surveys were conducted on a cross section of people that include dog
owners and non-dog owners and households that had rabies cases and those that did
not. The sampling that was described does not make it clear who was surveyed and
therefore it is not possible to have confidence in this statement. In the Limitation
section of the discussion the authors mention that the 263 mentioned are dog bite
cases. That would mean that the implications for these results can only be applied
to people who have been bitten by a dog, this needs to be stated clearly. Also, in
the author summary it is mentioned there are 938 dog bites, how does that relate to
the 263 mentioned here? 

Detailed comments:

Author summary:

• Include Zimbabwe in the description of the study area in the author summary.

• It is surprising you are able to get a significant result in a difference in
proportions when you only have 4 positive cases. Please describe this more clearly. 

Introduction:

• Rabies has no cure, but it is preventable with prophylaxis and this should be
mentioned. 

Methods:

• Change “It is fairly reliable” to “The rainfall is fairly reliable”

• Describe what the Likert Scale does and why it is used.

• Provide more detail on the sampling approaches, not just naming the methods. What
is the population from which each sample is taken (who could be selected)? How many
samples are taken? Be clear about the different target populations (all humans,
humans who reported dog bites and went to the hospital, etc.) 

Results:

• Provide the sample sizes along with the proportions for result.

• A table of the questionnaire results would be useful, similar to Table 1. 

• Provide the numerators and denominators for the tests on the likelihood of
contracting rabies.

Limitations:

• Not all dog bite cases could be reached and there was GPS uncertainty are in the
same sentence suggesting these are related points, why?

• Why do you believe the conclusions can be applied to all of Murewa District? This
is not clear. 

Conclusions:

• Can you quantify the risks based on the spatial maps produced?

Figures:

• Figure 1: the boarder colors for Ward 30 and Murehwa Wards are indistinguishable,
please correct this. 

• Figure 1: in the text it suggests that areas are not yet developed and the map is
referenced, but that information is not discernable from this map.

We cannot make any decision about publication until we have seen the revised
manuscript and your response to the reviewers' comments. Your revised manuscript is
also likely to be sent to reviewers for further evaluation.

Sincerely,

Amy J Davis, Ph.D.

Associate Editor

Sergio Recuenco

Deputy Editor

The authors present an examination of risk factors for rabies in Ward 30, Murewa
District, Zimbabwe. They identify risks associated with low awareness of rabies
risks and levels of unvaccinated dogs as primary concerns. The paper provides a nice
visual spatial analysis of dog bites, jackal presence, and rabies positive cases.
There are issues with clarity in the methods that impede the interpretation of the
results that needs to be addressed. 

The methods section could use more clarity. The methods state that “one family member
from each household that had a rabies case was selected for a proxy interview using
a questionnaire”. Were these households that had human rabies cases? Elsewhere, the
authors state that there were 4 rabies positive cases. So does this mean that only 4
households were surveyed? Where then does the 191 sample size come from. Please
clarify. Some terms that were used were also confusing and inconsistently used
(e.g., respondent, participant). In the results, the authors talk about 263
respondents. Where does this number come from? Only 191 were able to respond to the
questionnaire, so the 263 individuals should not be called respondents but perhaps
‘enrollees’. How were they enrolled? What questions were asked on the questionnaire?
A table of results from these questions would help.

Understanding how the people surveyed were selected is critical for readers to
understand the implications of these results. The authors state that “dog owners
were more likely to contract rabies as compared to non-dog owners”. This makes is
sound like surveys were conducted on a cross section of people that include dog
owners and non-dog owners and households that had rabies cases and those that did
not. The sampling that was described does not make it clear who was surveyed and
therefore it is not possible to have confidence in this statement. In the Limitation
section of the discussion the authors mention that the 263 mentioned are dog bite
cases. That would mean that the implications for these results can only be applied
to people who have been bitten by a dog, this needs to be stated clearly. Also, in
the author summary it is mentioned there are 938 dog bites, how does that relate to
the 263 mentioned here? 

Detailed comments:

Author summary:

• Include Zimbabwe in the description of the study area in the author summary.

• It is surprising you are able to get a significant result in a difference in
proportions when you only have 4 positive cases. Please describe this more clearly. 

Introduction:

• Rabies has no cure, but it is preventable with prophylaxis and this should be
mentioned. 

Methods:

• Change “It is fairly reliable” to “The rainfall is fairly reliable”

• Describe what the Likert Scale does and why it is used.

• Provide more detail on the sampling approaches, not just naming the methods. What
is the population from which each sample is taken (who could be selected)? How many
samples are taken? Be clear about the different target populations (all humans,
humans who reported dog bites and went to the hospital, etc.) 

Results:

• Provide the sample sizes along with the proportions for result.

• A table of the questionnaire results would be useful, similar to Table 1. 

• Provide the numerators and denominators for the tests on the likelihood of
contracting rabies.

Limitations:

• Not all dog bite cases could be reached and there was GPS uncertainty are in the
same sentence suggesting these are related points, why?

• Why do you believe the conclusions can be applied to all of Murewa District? This
is not clear. 

Conclusions:

• Can you quantify the risks based on the spatial maps produced?

Figures:

• Figure 1: the boarder colors for Ward 30 and Murehwa Wards are indistinguishable,
please correct this. 

• Figure 1: in the text it suggests that areas are not yet developed and the map is
referenced, but that information is not discernable from this map.

Reviewer's Responses to Questions

**Key Review Criteria Required for Acceptance?**

**Methods**

-Are the objectives of the study clearly articulated with a clear testable hypothesis
stated?

-Is the study design appropriate to address the stated objectives?

-Is the population clearly described and appropriate for the hypothesis being
tested?

-Is the sample size sufficient to ensure adequate power to address the hypothesis
being tested?

-Were correct statistical analysis used to support conclusions?

-Are there concerns about ethical or regulatory requirements being met?

Reviewer #1: See attachment

Reviewer #2: Objectives of the study is clear, but population description could be
improved to make clearer if only people in households with rabies cases, houses with
bitten people or houses in general were interviewed. 

Study design is appropriate but there is no mention of sample size or sampling
methodology in the strata of Ward 30. 

Statistical analysis in general are ok, however for dogs and jackals’ spatial
analysis, the procedures should be described with more detail.

I understand this study was in an outbreak context, however, IRB permits should be
mentioned, especially in children are included in the study population.

**Results**

-Does the analysis presented match the analysis plan?

-Are the results clearly and completely presented?

-Are the figures (Tables, Images) of sufficient quality for clarity?

Reviewer #1: Yes

Reviewer #2: Analysis matches the plan, however, not all variables described at the
beginning of results are mentioned or tested in the analysis section. Furthermore,
definitions of the categories of some variables are not provided. Results also may
need more order to improve readability. Dogs and jackal’s spatial analysis could be
better explained and connected. 

I would recommend presenting statistical analysis in a separate table. Descriptive
table 1 and all figures requires minor editions. Figure are of enough quality, but
they need bigger font sizes in the figure themselves and in the legends.

**Conclusions**

-Are the conclusions supported by the data presented?

-Are the limitations of analysis clearly described?

-Do the authors discuss how these data can be helpful to advance our understanding of
the topic under study?

-Is public health relevance addressed?

Reviewer #1: Conclusions supported by data. More details required on limitations.

Reviewer #2: Conclusions are supported but I considered that there are more
limitations that the one mentioned.

**Editorial and Data Presentation Modifications?**

Reviewer #1: Minor revision

Reviewer #2: I recommend to be careful with spaces after dots and some words
throughout the document and also format of numbers in English. An extra table to
present analysis and bigger font sizes in figures and their legends would help the
readers.

**Summary and General Comments**

Reviewer #1: See attachment

Reviewer #2: This study provides great information collected from an outbreak and
contributes to rabies endemic countries fight against rabies. Relevance of the study
even increases considering that in the study area there are multiple hosts that
allows rabies virus circulation for health authorities to be vigilant.

PLOS authors have the option to publish the peer review history of their article
(what does this mean?). If published, this will
include your full peer review and any attached files.

If you choose “no”, your identity will remain anonymous but your review may still be
made public.

**Do you want your identity to be public for this peer review?** For
information about this choice, including consent withdrawal, please see our
Privacy Policy.

Reviewer #1: Yes: Dr France Ncube

Reviewer #2: No

Figure Files:

Data Requirements:

Reproducibility:

comments on manuscript PNTD - 20- 00750.docx

---

## [Author Response · Author response to Decision Letter 0]

16 Dec 2020

to Reviewers 4 December for submission.docx
---

## [Editor Report · Decision Letter 1]

13 Jan 2021

Dear Dr macherera,

Thank you very much for submitting your manuscript "An assessment of risk factors for
contracting rabies in Ward 30, Murewa District, Zimbabwe" for consideration at PLOS
Neglected Tropical Diseases. As with all papers reviewed by the journal, your
manuscript was reviewed by members of the editorial board and by several independent
reviewers. The reviewers appreciated the attention to an important topic. Based on
the reviews, we are likely to accept this manuscript for publication, providing that
you modify the manuscript according to the review recommendations. 

Be consistent and clear throughout when referring to ‘cases’, be specific as to if
you mean dog bite cases, suspected rabies positive cases, or something else. 

In the abstract, define “the High Density Cluster”. 

In the study area section, what are “Growth Points” and why is this term
capitalized?

In Table 2, be clearer in the table caption or headers as to what the number column
represents (e.g., number of people who identified dogs as being a rabies reservoir). 

In Table 2, how are there 150 people who owned dogs, then in the next question only
49 people responded if their dog was vaccinated or not. 146 people having no pets
does not make sense with 150 saying they own dogs. Please clarify. 

In Table 3, give an example of how the questions were worded. The columns of ‘rabies’
and ‘no rabies’ are unclear. Were all people in the study asked these questions? 

The inclusion of Table 3 is helpful. The contingency tables within it are useful.
However, with only 4 suspected positive rabies cases the sample sizes are not
sufficient to conduct a chi-square tests, see McHugh 2013. Therefore, the chi-square
analyses should be removed from the manuscript. However, I think the value of the
information being shown is still there even without these tests. It is still
reasonable to look at odds ratios or risk ratios, as these are descriptive
statistics and not statistical tests that have sample size requirements. 

McHugh, M. L. 2013. The chi-square test of independence. Biochemia medica
23:143-149.

Sincerely,

Amy J Davis, Ph.D.

Associate Editor

Sergio Recuenco

Deputy Editor

Be consistent and clear throughout when referring to ‘cases’, be specific as to if
you mean dog bite cases, suspected rabies positive cases, or something else. 

In the abstract, define “the High Density Cluster”. 

In the study area section, what are “Growth Points” and why is this term
capitalized?

In Table 2, be clearer in the table caption or headers as to what the number column
represents (e.g., number of people who identified dogs as being a rabies reservoir). 

In Table 2, how are there 150 people who owned dogs, then in the next question only
49 people responded if their dog was vaccinated or not. 146 people having no pets
does not make sense with 150 saying they own dogs. Please clarify. 

In Table 3, give an example of how the questions were worded. The columns of ‘rabies’
and ‘no rabies’ are unclear. Were all people in the study asked these questions? 

The inclusion of Table 3 is helpful. The contingency tables within it are useful.
However, with only 4 suspected positive rabies cases the sample sizes are not
sufficient to conduct a chi-square tests, see McHugh 2013. Therefore, the chi-square
analyses should be removed from the manuscript. However, I think the value of the
information being shown is still there even without these tests. It is still
reasonable to look at odds ratios or risk ratios, as these are descriptive
statistics and not statistical tests that have sample size requirements. 

McHugh, M. L. 2013. The chi-square test of independence. Biochemia medica
23:143-149.
---

## [Author Response · Author response to Decision Letter 1]

15 Feb 2021

to reviwers comments 2021 FINAL.docx
---

## [Editor Report · Decision Letter 2]

10 Mar 2021

Dear Dr macherera,

We are pleased to inform you that your manuscript 'An assessment of risk factors for
contracting rabies in Ward 30, Murewa District, Zimbabwe' has been provisionally
accepted for publication in PLOS Neglected Tropical Diseases.

Best regards,

Amy J Davis, Ph.D.

Associate Editor

Sergio Recuenco

Deputy Editor

In Table 2, provide more detail about what ‘percentage’ means similar to the changes
made to the number of participants. Perhaps change to ‘percentage of positive
responses’.

---

## [Editor Report · Acceptance letter]

26 Mar 2021

Dear Dr Macherera,

We are delighted to inform you that your manuscript, "An assessment of risk factors
for contracting rabies in Ward 30, Murewa District, Zimbabwe," has been formally
accepted for publication in PLOS Neglected Tropical Diseases.

Best regards,

Shaden Kamhawi

co-Editor-in-Chief

Paul Brindley

co-Editor-in-Chief
